# A Machine Learning Framework Based on Extreme Gradient Boosting to Predict the Occurrence and Development of Infectious Diseases in Laying Hen Farms, Taking H9N2 as an Example

**DOI:** 10.3390/ani13091494

**Published:** 2023-04-27

**Authors:** Yu Liu, Yanrong Zhuang, Ligen Yu, Qifeng Li, Chunjiang Zhao, Rui Meng, Jun Zhu, Xiaoli Guo

**Affiliations:** 1Research Center of Information Technology, Beijing Academy of Agriculture and Forestry Sciences, Beijing 100097, China; 2National Innovation Center of Digital Technology in Animal Husbandry, Beijing 100097, China; 3College of Water Resources and Civil Engineering, China Agricultural University, Beijing 100083, China

**Keywords:** H9N2 status, laying hen house, machine learning, predictive performance

## Abstract

**Simple Summary:**

The H9N2 avian influenza virus has spread to the whole world and become one of the dominant subtype influenza viruses in chickens in China. H9N2 virus could not only result in great economic losses by reducing egg production but also serve as a gene vector to provide its gene segments to other emerging severe influenza A viruses to cause higher mortality and more serious consequences. Thus, developing models to predict H9N2 status should be given priority. Our main aim was to use the machine learning method (XGBoost classification algorithm) and regular production data (laying rate and mortality) to predict the occurrence and development of H9N2 in laying hen houses and evaluate their performance in disease prediction. Additionally, we assessed the working ability of the framework with different time frames to predict H9N2 status in advance within a 3-day period. We found that this framework could work well in prediction with high accuracy and sensitivity, and with more information introduced into the model, more “don’t care values” would be added into datasets to affect model performance by forming attribute noise. Besides, our study recommended efficient frameworks and models for H9N2 status prediction and provided practical potential uses in the livestock and poultry industry.

**Abstract:**

The H9N2 avian influenza virus has become one of the dominant subtypes of avian influenza virus in poultry and has been significantly harmful to chickens in China, with great economic losses in terms of reduced egg production or high mortality by co-infection with other pathogens. A prediction of H9N2 status based on easily available production data with high accuracy would be important and essential to prevent and control H9N2 outbreaks in advance. This study developed a machine learning framework based on the XGBoost classification algorithm using 3 months’ laying rates and mortalities collected from three H9N2-infected laying hen houses with complete onset cycles. A framework was developed to automatically predict the H9N2 status of individual house for future 3 days (H9N2 status + 0, H9N2 status + 1, H9N2 status + 2) with five time frames (day + 0, day − 1, day − 2, day − 3, day − 4). It had been proven that a high accuracy rate > 90%, a recall rate > 90%, a precision rate of >80%, and an area under the curve of the receiver operator characteristic ≥ 0.85 could be achieved with the prediction models. Models with day + 0 and day − 1 were highly recommended to predict H9N2 status + 0 and H9N2 status + 1 for the direct or auxiliary monitoring of its occurrence and development. Such a framework could provide new insights into predicting H9N2 outbreaks, and other practical potential applications to assist in disease monitor were also considerable.

## 1. Introduction

In recent years, various infectious diseases have posed risks to the livestock and poultry industry [1], such as avian influenza, foot and mouth disease (FMD), and classical swine fever. Infectious diseases are mostly caused by viruses, bacteria, fungi, etc. [2], which could rapidly outbreak and spread to greatly limit and test the epidemic prevention and response capacity of government and farmers. To date, infectious disease outbreaks have caused tremendous socioeconomic losses and public health consequences globally due to disease control measures [3], such as culling animals in endemic areas, temporary bans on the import and export of livestock products, and the limitation of livestock production and market activities [4].

For infectious disease management, the most appropriate strategy is accurate and rapid detection and control [5]. Researchers have developed some novel approaches of classifications based on association models to identify virus sequences, virus host range and zoonotic transmissible sequences to avoid pandemic or disastrous epidemics [6,7]. Infectious disease prediction has great value for initiating rapid responses and for the safety of the livestock and poultry industry. In order to well monitor animal health and diseases, many prediction models for infectious diseases have been developed in recent years [8] for early warning of individual animals, animal herds, and regions. Prediction models based on classical statistical and mathematical methods (e.g., logistic regression) have been widely demonstrated in epidemiological risk warning [9], which were usually considered as kind suitable techniques for identifying risk factors (e.g., lameness recognition in broiler [10]) rather than establishing outbreak prediction models [9,11]. There is increasing interest in trying to use machine learning methods for epidemic prediction [11,12], especially the tree-based machine learning methods, such as random forest (RF) and extreme gradient boosting (XGBoost), which have been widely used in predicting diseases with the advantages of high efficiency, and strong interpretability [13,14,15]. In the field of veterinary sciences, only a few studies focused on the prediction of disease outbreaks. RF was demonstrated to have good capability in predicting FMD outbreaks with the external risk factors as input, like purchasing, neighbors, vaccinations and close to the main road [12], and a knowledge map was maturely applied to diagnose livestock and poultry diseases with varied symptoms [16,17]. Obviously, additional time and labor are required to collect and count these input variables mentioned above in addition to the normal production data. It would be more convenient and beneficial for farmers and veterinarians to identify animal health abnormalities and rapidly respond if the regular production data could be used directly to reflect disease outbreaks, but it is still a gap. Meanwhile, the developed models just predicted the current outbreak status and were not forward-looking. Many studies have been using a time-series forecasting model to explain, evaluate and estimate the development trends in outbreak human diseases (e.g., COVID-19), such as the further values of outbreak disease cases, deaths, or transmission rates [18,19]. Besides, XGBoost was proved to have more outstanding prediction accuracy and generalization ability than RF, support vector machines (SVM), k-nearest neighbor (KNN), and back propagation neural network (BPNN) in predicting the risk and outbreaks of diabetes [20], dermatomyositis [21], COVID-19 [14,15], wheat stripe rust [22], and other diseases of human and plants, due to the superiority of its architecture, which also provided a new idea and algorithm for risk/outbreak identification, early intervention, and practical application in animal diseases.

Different infectious diseases do not only show different symptoms in pathological changes but also show different effects on production performance indicators due to differences in virulence, adaptability and replication; even different strains of the same disease can have different clinical manifestations [23]. For example, there were significant differences in the survival percentage and weight-changing trends of broilers infected with HN10PY01 and NX0101 of avian leukosis virus-J field strains [23]. One study showed the survival rate was 30%, 50%, and 90% in broilers inoculated with *Ornithobacterium rhinotracheale* (ORT) + H9N2 avian influenza virus, ORT and H9N2 virus alone, respectively [24]. These indicators mentioned above are the regular ones recorded every day in most commercial farms, which supports the use of regular production data to reflect different diseases. Even though some different diseases show similarities in production data, the prediction of infectious disease outbreaks with regular production data will also be beneficial to help governments and farmers rapidly respond to screening and control infectious diseases.

Generally, effective detection and monitoring are necessary for infectious disease management [25]. This study used the data from laying hens infected with the H9N2 virus as an example, trying to use a machine learning method (XGBoost classification algorithm) and regular production data (laying rate and mortality) to predict the occurrence and development of H9N2 and prove the prediction performance of the framework and XGBoost models to provide an efficient method for infectious diseases early warning in livestock and poultry farms.

## 2. Materials and Methods

Avian Influenza caused by H9N2 avian influenza virus (“H9N2 Avian Influenza” referred to as “H9N2” below) can pose a serious threat to both the poultry industry and public health safety, and an effective and rapid diagnostic method for H9N2 occurrence and development is urgently needed for prompt prevention and control.

### 2.1. H9N2 Cases and Data Collection

This study collected information on H9N2 occurrence and development and correlated production data from a large-scale concentrated laying hen farm located in Beijing. Trained and experienced veterinarians, who routinely conduct disease identification and abnormity diagnosing, helped to confirm the H9N2 cases (including the start and end of this epidemic) with their experiences and the results of the avian influenza virus antigen detection card based on the colloidal gold method.

Data were collected from 3 infected laying hen houses every day from the period of October to December 2012, covering the whole onset cycle of each outbreak. The numbers of days of having H9N2 and non-H9N2 were 37 and 88, with a case-to-control ratio of approximately 1:2, fitting the typically recommended ratio of observational studies of disease incidence [26]. Variables collected in this study are listed in Table 1. Temperature and relative humidity were automatically measured by sensors and recorded in the environment management system. Date, age, laying rate and mortality were automatically generated, calculated and recorded by the management system. The number of abnormal laying hens, the qualified rate of immune antibodies against highly pathogenic avian influenza (HPAI) (QR_immu_), and the treatment for H9N2 infection were confirmed, recorded, and provided by a veterinarian. Veterinarians treated the laying hens with drugs from the day that the first H9N2-infected bird was detected until a non-H9N2-infected bird was confirmed.

### 2.2. Xgboost Algorithm

XGBoost is 1 of the latest tree-boosting algorithms based on a novel sparsity-aware algorithm and weighted quantile sketch to develop a “strong” learner using all prediction results of “weak” learners [27,28] with great success in terms of both performance and speed. As a tree-based model, the XGBoost classification algorithm changes the score of leaves to a class tag and finally outputs the class tag with the highest proportion of allocated leaves. In this study, the XGBoost classification model was built to predict H9N2 status to reflect the occurrence and development process of this epidemic, and Figure 1 uses an example to illustrate the progress of the XGBoost classification model using 2 trees, each with 2 depths to predict the status. In fact, there are always more than 2 trees and 2 depths in model training. H9N2 status was dealt with binary problems with the labels of 0 and 1, of which “0” meant “no H9N2 case”, and “1” meant “having H9N2 case”. If the sum of the function leaf scores was greater than 0.5, it was judged to belong to 1; otherwise, it belonged to 0. Features including laying rate, mortality and time frame features, which were directly and closely related to H9N2 oncomes, were used as the input.

Hyperparameters, such as the regularization parameters in XGBoost, can greatly affect the performance of a model [29]. In this study, Grid Search Cross-Validation exhaustive parameter search technique with fivefold cross-validation was applied to search the optimal hyperparameters involved in the performance of the models, which included learning_rate, n_estimators, max_depth, min_child_weight, gamma, subsample, and colsample_bytree, with the search spaces of 0.001~0.2, 0~100, 1~15, 1~15, 0~1, 0.7~0.9, and 0.7~0.9, respectively. The optimized hyperparameters were determined by the maximum value of the accuracy. Then, the importance of these features in building and boosting the tree-based XGBoost classification model was calculated for model interpretation [30].

### 2.3. Framework Design

All of the XGBoost classification algorithm running, hyperparameters tuning, and statistical analysis were conducted using Python software 3.6 and the scikit-learn toolkit. The examples of Python codes and collated datasets were provided as Appendix A.

The overall workflow for the prediction task is given in Figure 2. In order to prove the model’s good application ability, the collected datasets from 2 infected laying hen houses were used as a training set (about 64% of all data), and datasets from the 3rd infected laying hen house were used as a testing set (about 36% of all data). Fivefold cross-validation was conducted in each model training process to evaluate the performance of such a framework under practical farm management conditions. The hyperparameters were tuned only utilizing the training set. The input factors included the laying rate with time frames and the mortality with time frames, and the output value was H9N2 status. After training XGBoost models, the accuracy, recall rate, precision rate and receiver operator characteristic (ROC) curve of each model were calculated to evaluate the framework performance.

Laying hens inevitably suffer different diseases or breeding equipment failures in the normal breeding process, which will affect the normal laying rate and mortality changing regulars. In order to avoid the impact of outliers in long-term production data on the classification performance of the model and to reduce the data requirement for small sample events such as infectious disease outbreaks, this study tried to investigate the suitable time frame using for H9N2 status confirming. Given that H9N2 can cause sudden changes in production performance, choosing long-term data is not obviously helpful for the prediction of disease outbreaks; this study set 5 time frames rolling windows of today (day + 0), the previous 1 day (day − 1), the previous 2 days (day − 2), the previous 3 days (day − 3), and the previous 4 days (day − 4) to study the suitable time frame for H9N2 status confirming. In each time frame case, 3 XGBoost models were built to predict coupled future 3 days’ values, of which model #1 would predict today’s status (i.e., H9N2 status + 0), model #2 would predict the next day’s status (i.e., H9N2 status + 1), and model #3 would predict the future 2nd day’s status (i.e., H9N2 status + 2). The H9N2 status was defined into 2 categories, “no” (marked as 0) and “yes” (marked as 1), of which “no” meant no H9N2 case, and “yes” meant having an H9N2 case. Due to each prediction being targeted on a 3-day period, a time series 3-day rolling window was applied to create input and output variables in train or test datasets, as shown in Figure 3.

### 2.4. Evaluation Criteria

There were 4 generally used criteria calculated to evaluate and compare the prediction capacity of developed models with different time frames. This study defined the 0 status (no H9N2 case) as negative and the 1 status (having H9N2 case) as positive. The agreement of the results between the actual and predicted status was described by the number of true positives (*TP*), true negatives (*TN*), false positives (*FP*), and false negatives (*FN*), of which the sum of *TP*, *TN*, *FP* and *FN* was the total number of observed days in 3rd laying hen house (*N*). Accuracy rate, recall rate, precision rate, and the area under the curve (AUC) of the ROC were used to assess the performances of the XGBoost models [12]:(1)Accuracy rate (ACC)
(1)$$ACC=\frac{TP+TN}{N}\times 100\%$$

(2)Recall rate (RR)


(2)
$$RR=\frac{TP}{TP+FN}\times 100\%$$


(3)Precision rate (PR)


(3)
$$PR=\frac{TP}{TP+FP}\times 100\%$$


AUC of ROC

AUC is an indicator of the discriminative or predictive ability of prediction models. The prediction ability could be generally judged based on the AUC values: AUC = 0.5, no discrimination; 0.5 < AUC < 0.6, poor discrimination; 0.6 ≤ AUC < 0.7, fair discrimination; 0.7 ≤ AUC < 0.8, acceptable discrimination; 0.8 ≤ AUC < 0.9, excellent discrimination; and AUC ≥ 0.9, outstanding discrimination [31]. A higher AUC value means the model prediction performance is better.

## 3. Results

### 3.1. Descriptive Statistic

The descriptive statistics associated with the variables about the information of H9N2 occurrence and development and correlated production data are shown in Table 2. The age range of studied birds was 161–266 days, with an average laying rate of 93.3% and an average mortality of 0.03% during the normal production period. Their average age was around 27 weeks, which was in the peak laying period [32]. During the period of data collection, the temperature ranges of houses 1 and 2 were similar, with T_max_ values from 3 to 24 °C and T_min_ values from −10 to 9 °C; the temperature of house 3 was significantly lower than the other 2 houses (*p* < 0.05). The average RH range of 3 houses was from 35 to 65%. With the QR_immu_ above 80%, the extreme values of laying rate, mortality, and N_ab-hens_ were 84.1%, 0.20%, and 265 hens, respectively, after the hens were infected with H9N2.

The data from house 1 was taken as an example to reflect the basic changing of environment and production variables in Figure 4. It was clear that the temperature showed a slow downward trend; RH fluctuated regularly and was stable in total. Before the H9N2 outbreak, the laying rate increased gradually and stabilized at 96% according to the normal growth curve of laying hens. Meanwhile, the mortality regularly changed to around 0.02%. After the H9N2 outbreak, both the laying rate and mortality were sharply affected. Interestingly, there were slight abnormal changes in laying rate and mortality occurring 1 or 2 days before the outbreak, which could be mined to be the portends of the H9N2 outbreak.

### 3.2. Predictive Performance Comparison

Table 3 presents the comparative results of the model performances with different time frames to predict 3 future days’ H9N2 status. All the prediction models had accuracy rates greater than 80% with high recall rates, which meant the high sensitivity of these models. It could be seen day + 0 and day − 1 classification models had the same accuracy and sensitivity in predicting H9N2 status + 0, +1 and +2. With the accumulated days used as input data increasing, all of the ACC, RR and PR fluctuated and decreased. The XGBoost classification models predicted the likelihood of the H9N2 status + 0 most precisely since they revealed the highest PRs. ACC and RR of models for H9N2 status + 0 kept stable at 92.31% with the accumulated days from today to the previous 3 days, then they decreased with the input days increasing. It was worth noting that the performance of the day − 2 model was the lowest among all of the models to predict H9N2 status + 1; other models had the same ACC and RR. Meanwhile, for H9N2 status + 2 prediction, all of the models had the same performances of ACC and RR. It was clear that when the models predicted the H9N2 status of more future days, the performances were worse in general. The performances of the models of days + 0, −1, −2, and −3 to predict H9N2 status + 0 were the best, with an ACC and RR of 92.31%, and H9N2 status + 1 was the next.

A higher AUC value shows a better prediction performance of the evaluated model. Analyzing Figure 5, the AUC values of most models were higher than 0.90, indicating their outstanding discrimination. Regarding the AUC values, similar to the results from Table 3, the more days the models predicted, the worse the performances were. The performance of models predicting H9N2 status + 0 was the best, with all the AUC values higher than 0.90, followed by H9N2 status + 1 and H9N2 status + 2. When the input data came from more days, the AUC values would become worse. ACC and RR of models for H9N2 status + 0 kept stable at 92.31% with the accumulated days from today to the previous 3 days, then they decreased with the input days increasing. Models of day + 0 and day − 1 to predict the H9N2 status of different future days had similar great performances, and the highest AUC value was found in the day − 1 model for H9N2 status + 0 with an AUC of 0.98.

### 3.3. Feature Importance

Due to the best performances of models to predict H9N2 status + 0, Figure 6 takes H9N2 status + 0 prediction models as examples to analyze the feature importance ranking, which refers to the contributions of individual input variables to the performance of the model. In most of these models, laying rate and mortality were identified as playing different roles in building models, except the day + 2 model for H9N2 status prediction, where only laying rates were selected in building the model. This indicated the variations in laying production and mortality between different days were the key driver in training models. The results revealed that laying rate was the more important feature in predicting H9N2 status than mortality. If the production data only from today and the previous day were used as input, the laying rate + 0 showed its great importance in prediction models. While, with the date from more accumulated days inputting, laying rate + 0 seemed not as important as before, the laying rates of the previous 1 and more days played the key role in prediction. Similarly to mortality, the importance of mortality from today and the adjacent days was lower and lower. For instance, the mortality from different days had similar importance and was ranked last.

## 4. Discussion

In the present study, the XGBoost classification algorithm was applied to H9N2-related data to develop models designed to predict the H9N2 status among laying hen houses using temporal variables and production variables as predictors. The predictive performances of XGBoost models were evaluated and compared.

### 4.1. Relationship between Environment and Production Variables and H9N2 Status

Studies proved low temperatures would help avian influenza viruses survive in the environment to facilitate viruses’ introduction and outbreaks [33]. Meanwhile, the low temperature would decrease the hens’ immunity, which could potentially cause the H9N2 outbreak in the laying hen house. As shown in Table 2 and Figure 4, the temperature gradually dropped even to −14 °C, which was far away from the optimum temperature (thermoneutral zone, 19–22 °C) for laying hens [34]. And it could be proved by the correlation analysis in Table 4.

Since the hens were infected, H9N2 would spread to the whole house, and H9N2 had risen to be the most prevalent subtype of avian influenza virus in China, causing great economic losses because of the reduced egg production [35,36]. Clearly, after the H9N2 outbreak, the laying rate sharply decreased and was significantly and negatively correlated with H9N2 positive status (*p* < 0.01). Besides, H9N2 would cause high mortality associated with co-infection with other pathogens [37], such as H5N2 [38], H6N1 [39], H7N7 [38], H7N9 [40], and H10N8 [41]. H9N2 could also cause mortality due to the pathogenicity itself or the enhancement of other disease responses. In Korea, when layers were infected with H9N2, the mortality was about 30%, with reduced egg production [42]. In our study, all of the laying hens were vaccinated with immune antibodies against HPAI, with a qualification rate of above 80%. This H9N2 epidemic just caused a small increase in mortality but still significantly and positively affected the mortality (*p* < 0.01).

Furthermore, 1 or 2 days before the H9N2 had been detected, the laying rate and mortality changed abnormally compared with the data from the days before. In this stage, H9N2 was not detected, which might be because of its low concentration, or because it caused such little impact on production to fail to raise the alarm among farmers and veterinarians. For epidemics, a delay of 1 or 2 days can lead to a wider impact on animal health and performance, especially the H9N2 can serve as vehicles by donating their gene segments to other influenza A viruses [37] to cause more serious consequences. Thus, developing prediction models to control the circulation of H9N2 should be paid more attention to, and the laying rate and mortality as the priority corresponding factors could be used as input variables for H9N2 status prediction models.

### 4.2. Framework Performance and Model Interpretation

Several studies have reported computational fluid dynamics modeling, air dispersion modeling, and transmission network modeling on HPAI using past outbreaks of infectious viral animal diseases to determine the contribution of airborne transmission towards HPAI spread [43] during its outbreaks. As one of the important subtypes of the avian influenza virus, H9N2 could donate gene segments to other HPAI viruses. Thus, developing models to predict and control the circulation of H9N2 should be given priority.

Different from analytical models used in finding the relationship between different variables [44], the epidemic-based predictive model is focused on reliable and accurate prediction under real production. This study evaluated the XGBoost classification models with different time frames for predicting the H9N2 status of several future days. This setting ensured the model’s development and future prediction were only dependent on historical information. By analyzing the performance criteria, the trend and fluctuation of ACC, RR, PR, and AUC can be seen in Table 3 and Figure 5, and it was found that the accuracy, sensitivity, and AUC values of the models to predict H9N2 status + 0 were higher than those to predict H9N2 status + 1 and +2, the drop of accuracy was 7.69% with the prediction moving forward from 0th to 2nd day. The overall accuracy rate and recall rate of this framework to predict H9N2 status + 0 with the time frame of day + 0, day − 1, day − 2 and day − 3 were demonstrated as reliable toward 92% of the predictions, and the framework to predict H9N2 status + 1 with the time frame of day + 0 and day − 1 still worked well. Besides, the data captured herein represent the before/after H9N2 occurrence and development in concentrated and HPAI-immunized laying hen houses; in order to prove the good applications of our results, data from two of the infected houses were used in model training, and data of the third house was used in model testing. Furthermore, it would be beneficial to apply the models to larger datasets collected from other endemic areas in recent years.

Once the animal suffers from a disease, the production performance usually changes suddenly. Unlike time series cross-validation models, this study chose five time frames with a fixed number of days rather than the accumulated days to predict H9N2 status. Compared with ACC, RR, PR, and AUC of different models, the models of time frames with more days generally showed worse and fluctuant performances in predicting H9N2 status + 0 and +1, which might be due to more additional and useless information introduced into the model training. For the prediction of a binary outcome, imbalanced classification often occurs when many outputs belong to one of two classes [45]. As shown in Figure 6, the laying rate and mortality of day − 3 and day − 4 played small roles in each model, and some features were even negligible. This finding was supported by the study indicating that the more information introduced into the model, the more “don’t care values” would be added into datasets, which was easy to form attribute noise affect model performance [46]. The feature with a higher value indicates its greater power to explain the variation of the output and its key role in training the model [47]. Laying rate − 1 and laying rate − 2 ranking the first two were the top two important predictors in models with day − 2, −3, −4 instead of laying rate + 0 as the key predictor in models with day + 0 and day − 1, illustrating laying rate + 0 had the significant and stronger explanation for epidemic status. Similarly, for H9N2 status + 2, the ACC and RR of all models were 84.62%, and their PRs were 73.33% because their first 2 important features were laying rate + 0 and laying rate − 1 to show their similar explanation ability except day + 0 model, which only had 2 features ranking by laying rate + 0 and mortality + 0 (Appendix A).

Furthermore, in XGBoost classification models, the laying rate had higher importance than mortality features; this was consistent with the conclusions of the impact of H9N2 on poultry production [36], where the H9N2 mainly resulted in directly reducing egg production or high indirect mortality associated with the co-infection with other pathogens. In this study, XGBoost classification models of day + 0 and day − 1 to predict H9N2 status + 0 and H9N2 status + 1 were recommended for H9N2 occurrence and development predicting. Besides, there were a lot of machine learning methods used in animal disease predictions [12,48], and Gradient-Boosted Tree (GBT) was proven to be the most accurate model (accuracy of 84.9%) than Deep Learning, Decision Tree, and RF in predicting sub-clinical bovine mastitis [48]. XGBoost, as a quick implementation of GBT [49], was proved to have more outstanding prediction accuracy and generalization ability than RF, support vector machines (SVM), k-nearest neighbor (KNN), and back propagation neural network (BPNN) in predicting the risk and outbreaks of human diseases [14,15,20,21], due to its parallel processing [30], handle and capture missing values internally [50]. The results of our study also showed good application and performance in animal infectious disease predictions.

### 4.3. Potential Applications

This study provided an acceptable and reliable framework to predict H9N2 status from +0th to +2nd days with a high accuracy rate, recall rate and AUC within the cross-validation. Different from the prediction generated by traditional production curves [51,52], this framework was more dynamic and individual-house-oriented, which could support many innovative applications. Basically, authorities or veterinarians may use XGBoost classification models to communicate the risk of H9N2 occurrence and development to farmers and stakeholders, and the warning rule could be described as Table 5 or revised based on the veterinary’s criteria. The status of two consecutive days can be used to reflect the occurrence and development of H9N2 and give the corresponding warning level to remind and assist authorities or veterinarians in diagnosing and controlling the disease. Therefore, precise treatment can be given, such as medicating, culling or temporary bans [4]. The same framework could also be applied to early warning and auxiliary diagnosis of similar diseases that have impacts on production performance or other production traits responded in advance, such as infectious laryngotracheitis [53] and fowlpox [54]. Alternatively, the applications of these XGBoost classification models can be integrated with computer-based animal epidemic surveillance systems (e.g., the web-based FMD prediction system of Thailand [12]) or farm management systems as an additional application for H9N2 status prediction. The likelihood of an H9N2 outbreak for a particular laying hen house can be determined through these models whenever the new real-time data are recorded in the management system based on the judgment rule listed in Table 5 and through the color of the indicator on the screen to intuitively reflect H9N2 status in laying hen house.

At present, Python software used in this study is free and open access, which has the ability to handle different types of data, from small to large datasets. Moreover, many open-source machine learning platforms are available and easy to operate only if the users have basic computer or statistical programming knowledge [11].

## 5. Conclusions

This study developed a framework using XGBoost classification models and demonstrated that their abilities with different time frames were capable of predicting H9N2 status in individual laying hen houses in advance within a 3-day period. H9N2 status prediction models worked well with accuracy and sensitivity ranging from excellent to outstanding, of which an accuracy rate > 90%, a recall rate > 90%, a precision rate > 80%, and an AUC ≥ 0.85 could be achieved. Among them, the models of day + 0 and day − 1 were highly recommended to predict H9N2 status + 0 and H9N2 status + 1 for direct or auxiliary monitoring of the occurrence and development of H9N2. Authorities and veterinarians may consider applying the XGBoost classification algorithm to the currently used computer-based animal epidemic surveillance system or farm management system to enhance its early warning capability. Furthermore, the framework utilized in this study may also be considered for application to data from other infectious diseases, which have similar impacts on production performances.

## Figures and Tables

**Figure 1 animals-13-01494-f001:**
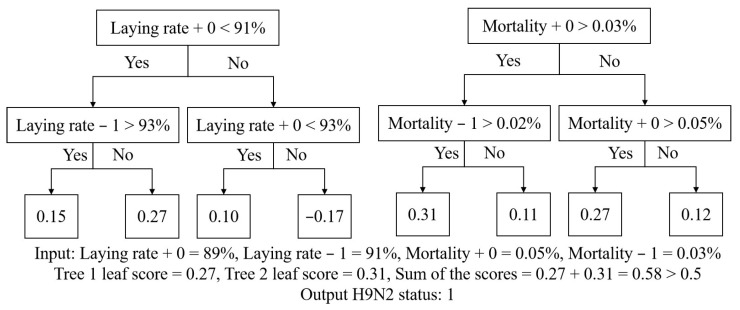
Illustration of XGBoost classification model using 2 trees, each with 2 depths to predict H9N2 status, an example. Laying rate + 0 = laying rate of today, laying rate − 1 = laying rate of yesterday, mortality + 0 = mortality of today, mortality − 1 = mortality of yesterday.

**Figure 2 animals-13-01494-f002:**
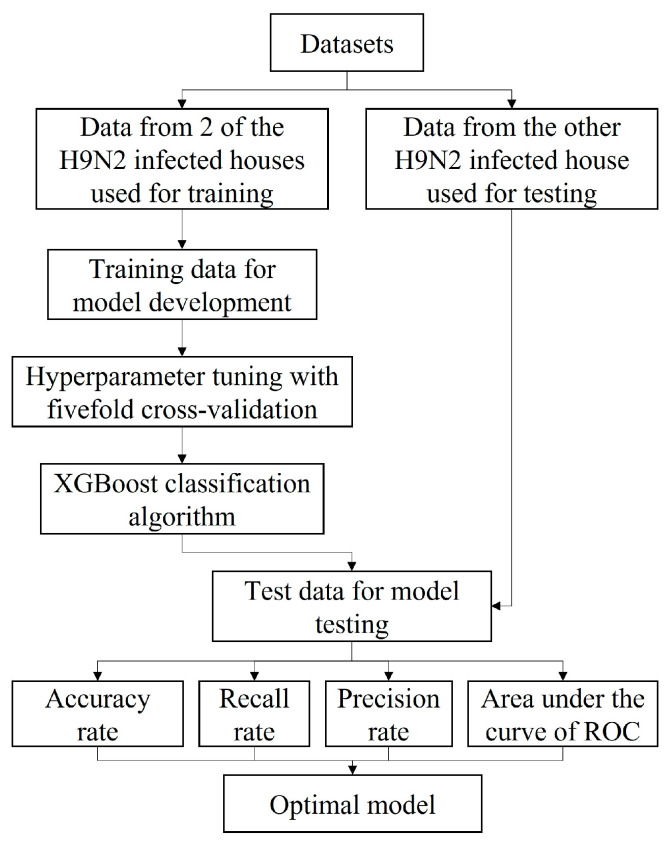
Overall workflow of model development for H9N2 status prediction. ROC, receiver operator characteristic curve.

**Figure 3 animals-13-01494-f003:**
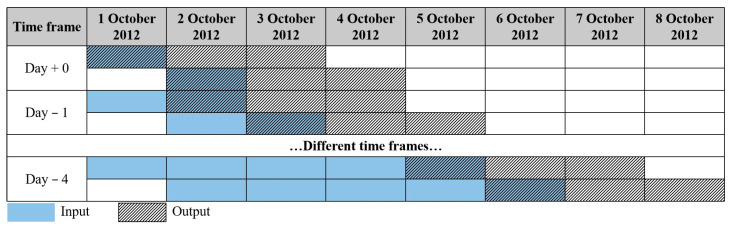
Examples of different time frames to simulate the execution of the XGBoost classification model framework. +0 = today, −1 = previous 1 day, −2 = previous 2 days, −3 = previous 3 days, and −4 = previous 4 days.

**Figure 4 animals-13-01494-f004:**
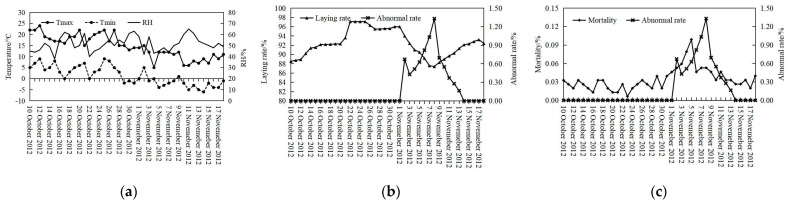
Taking house 1 as an example to show the changes in environment and production variables during the data collection period: (**a**) T_max_, T_min_, and RH; (**b**) Laying rate and abnormal rate; (**c**) Mortality and abnormal rate.

**Figure 5 animals-13-01494-f005:**
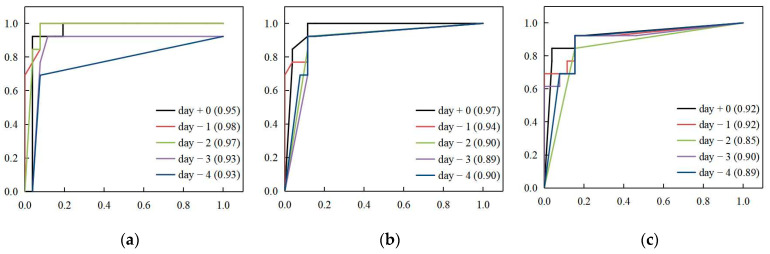
Receiver operator characteristic (ROC) curve from XGBoost models to predict H9N2 status of today (**a**), next day (**b**), and the future 2nd day (**c**) in laying hen houses with time frames of day + 0, day − 1, day − 2, day − 3, and day − 4. The value enclosed in parentheses meant the area under the curve. +0 = today, −1 = previous 1 day, −2 = previous 2 days, −3 = previous 3 days, and −4 = previous 4 days.

**Figure 6 animals-13-01494-f006:**
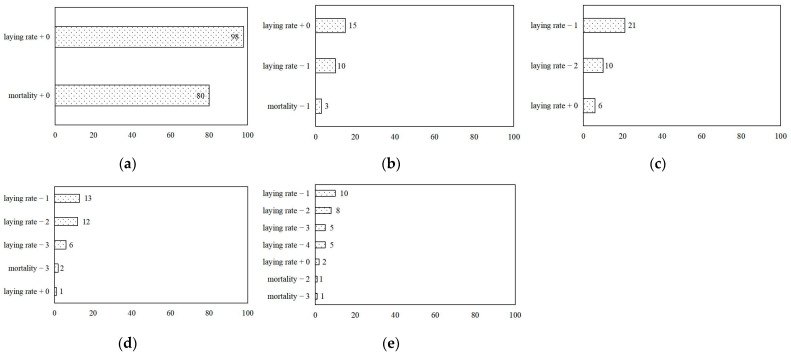
Feature importance for H9N2 status + 0 prediction in laying hen houses from the XGBoost classification models with time frames of day + 0 (**a**), day − 1 (**b**), day − 2 (**c**), day − 3 (**d**), and day − 4 (**e**). +0 = today, −1 = previous 1 day, −2 = previous 2 days, −3 = previous 3 days, and −4 = previous 4 days.

**Table 1 animals-13-01494-t001:** Variables about the information of H9N2 occurrence and development and correlated production data.

Variables	Variables
Date	Laying rate/%
Age/days	Mortality/%
Maximum temperature throughout the day (T_max_)/°C	Number of abnormal laying hens (N_ab-hens_)
Minimum temperature throughout the day (T_min_)/°C	Qualified rate of immunization (QR_immu_)/%
Average relative humidity (RH)/%	Treatment for H9N2 infection

**Table 2 animals-13-01494-t002:** Variable description of the overall data in the 3 H9N2 infected laying hen houses.

Variable		Unit	Max	Mean	Min	SD
House 1	Age	days	199	-	161	-
	T_max_	°C	24	15	5	5
	T_min_	°C	9	1	−6	4
	RH	%	65	52	38	7
	Laying rate	%	97.1	92.5	87.4	2.9
	Mortality	%	0.10	0.03	0.01	0.02
	N_ab-hens_	hens	201	-	0	-
	QR_immu_	%	90	90	90	0
House 2	Age	days	258	-	218	-
	T_max_	°C	22	11	3	5
	T_min_	°C	9	−2	−10	4
	RH	%	65	49	35	8
	Laying rate	%	94.3	92.4	86.1	2.3
	Mortality	%	0.20	0.04	0.01	0.03
	N_ab-hens_	hens	265	-	0	-
	QR_immu_	%	80	80	80	0
House 3	Age	days	266	-	222	-
	T_max_	°C	12	4	−4	5
	T_min_	°C	1	−6	−14	4
	RH	%	65	46	35	7
	Laying rate	%	93.4	91.4	84.1	2.7
	Mortality	%	0.13	0.05	0.01	0.03
	N_ab-hens_	hens	214	-	0	-
	QR_immu_	%	80	80	80	0

**Table 3 animals-13-01494-t003:** Model performance parameters, including accuracy (ACC), recall rate (RR) and precision rate (PR) from 15 XGBoost models with 5 time frames of today (day + 0), previous 1 day (day − 1), previous 2 days (day − 2), previous 3 days (day − 3), and previous 4 days (day − 4) to predict future 3 days’ H9N2 status (+0, +1, +2).

Criteria	Predicted Future Days	Day + 0	Day − 1	Day − 2	Day − 3	Day − 4
ACC/%	H9N2 status + 0	92.31	92.31	92.31	92.31	89.74
	H9N2 status + 1	89.74	89.74	87.18	89.74	89.74
	H9N2 status + 2	84.62	84.62	84.62	84.62	84.62
RR/%	H9N2 status + 0	92.31	92.31	92.31	92.31	84.62
	H9N2 status + 1	92.31	92.31	84.62	92.31	92.31
	H9N2 status + 2	84.62	84.62	84.62	84.62	84.62
PR/%	H9N2 status + 0	85.71	85.71	85.71	85.71	84.62
	H9N2 status + 1	80.00	80.00	78.57	80.00	80.00
	H9N2 status + 2	73.33	73.33	73.33	73.33	73.33

**Table 4 animals-13-01494-t004:** Correlation analysis of environment and production variables and H9N2 status (yes or no) in 3 laying houses.

Laying Hen House	T_max_	T_min_	RH	Laying Rate	Mortality
House 1	−0.61 **	−0.45 **	−0.01	−0.58 **	0.67 **
House 2	−0.30	−0.36 *	−0.31 *	−0.86 **	0.66 **
House 3	−0.67 **	−0.52 **	−0.21	−0.87 **	0.83 **

* Significant difference at 0.05 level. ** Significant difference at 0.01 level.

**Table 5 animals-13-01494-t005:** Warning rule of H9N2 status using XGBoost classification models.

Predicted H9N2 Status		Warning Level	Meaning	H9N2 Development
H9N2 Status + 0	H9N2 Status + 1			
0	0	Green	Safe	None
0	1	Yellow	Low warning	Start
1	1	Red	High warning	Development
1	0	Yellow	Low warning	Nearly finish

Note: +0 = today, +1 = future 1st day.

## Data Availability

The data presented in this study are available on request from the corresponding author.

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
