# Peer review of "A Machine Learning Framework Based on Extreme Gradient Boosting to Predict the Occurrence and Development of Infectious Diseases in Laying Hen Farms, Taking H9N2 as an Example"

_animals, 2023, doi:10.3390/ani13091494_

Round 1
Reviewer 1 Report
Although the manuscript is simple and easy to read, it lacks novelty, so it needs to improve significantly to meet scientific standards. Here are some major comments that the authors have to address. Also, it is recommended to correct your manuscript minutely for minor mistakes.
1. There is number of research in disease prediction using machine learning models. But the author reviewed a few of them related to their work. Please review some more relevant papers, find out the knowledge gaps in existing research and justify your work with the knowledge gap.
2. In line 68, the authors mentioned that the random forest demonstrated superior performance in veterinary science, but why the author chose the XGBoost model? It is recommended to use other models as well and select the best one.
3. The data collection period is mentioned from October to December 2012. Why did the author use such old data? And what is the data collection period? The author mentioned in the abstract they used 5 days of data, which is contradicted by the information mentioned in the material and method section.
4. In the variables, why the authors chose minimum and maximum temperature? Why not average temperature? There is only one relative humidity variable. Is it average or min, or max RH?
1 5. Since the dataset used for training and testing is very old, please validate your model with the latest data and explain the model performance in a new dataset.
Author Response
Dear Academic Editor and Reviewer,
Thank you very much for reviewing our manuscript entitled "A machine learning framework to predict the occurrence and development of infectious diseases in laying hen farms-taking H9N2 as an example". We have carefully revised the paper following your comments. Please see the attachment.

Reviewer 2 Report
In this paper, authors well described and investigated a method to predict the occurrence and development of H9N2 in laying hen farms using machine learning framework, which also has reference significance for the prediction, prevention and control of other infectious diseases. The results achieved are important in the point of the severe influence and losses in laying hen farm caused by H9N2 virus. At the same time, I really appreciated the effort of the authors to provide a ready to use protocol that can be easily followed by "on field" farmers.
However, there are few issues that should be addressed before publication:
My comments are listed below for the authors to consider:
1. My only minor concern is the improvement of the paper about the precautions that could be applied to predict the occurrence and development of H9N2 by considering the obtained results. How will this study affect the "laying hen houses" in the future? It might be better to add more sentences on this subject.
2. The main contribution should be emphasized in the “Introduction” section.
3. The title and content of the table must be self-explanatory, and all abbreviations should be explained in the footer of the table. For example, the “ROC” in Figure 1. Please check the full manuscript.
4. Line 99-102: “referred as H9N2 below” would make confuse with “H9N2” below, where the “H9N2” meant “H9N2 avian influenza”. Here, the “H9N2 avian influenza virus” should referred as “H9N2 virus”, please correct.
5. Line 145-146: Why don’t you use the commonly used method of data partitioning, such as 80:20 or 70:30, in this study?
6. In table 3, the values of the same variable should keep the same significant digits, like Tmax, Tmin.
7. Line 369, please list some diseases that have potential applications using this machine learning framework to express the meaning of the research more intuitively.
Author Response

(The authors gave the same response as above.)

Reviewer 3 Report
This manuscript is of great interest to the poultry industry, as infectious diseases among laying hens may lead to significant economic losses. This work applied the XGBoost algorithm and addressed a classification problem, which has novelty in terms of applying machine learning algorithms. However, several sections in the manuscript are less informative that require more details, and the workflow and validation results could be improved. My major comments are below.
Line 22: I recommend changing ‘prove’ to ‘validate’ or ‘evaluate’.
Introduction: This section lacks information about the reason to use mortality rate and laying rate as predictors to detect the H9N2 disease. In addition, there is no justification for the reason to use XGBoost algorithm in this study.
Section 2.2: Technical terms such as learners, leaves, and learning rate were never defined before in this manuscript. It would be challenging for the readers of the journal to fully understand the XGBoost algorithm. At minimum, the authors should include the illustration of the algorithm in the method section or in the supplementary materials e.g., graphs, formulas, or diagrams. In addition, it is not clear how the predictor variables are used as input to the XGBoost algorithm.
Line 151: what was the reason for not computing the precision rate? Precision is an important metric if the model will be in use for production or future prediction.
Line 273: I suggest the authors elaborate on the word ‘compared’. Was there a reference algorithm considered as the gold standard in this domain? Was the reference model used and compared with XGBoost in this study?
Lines 328-330 and Figure 1: the authors may be careful when using the term ‘generalizability’. In this study, data from three housing units were collected. Accordingly, three-fold cross-validation by housing units could be performed i.e. there can be three training-testing scenarios:
1) Training: Data from Houses #1 and #2; Testing: Data from House #3
2) Training: Data from Houses #1 and #3; Testing: Data from House #2
3) Training: Data from Houses #2 and #3; Testing: Data from House #1
The analytical framework was only applied in Scenario 1), while the validation results from 2) and 3) remained unknown. I would encourage the authors to either redefine ‘generalizability’ in the context or train and test the model under Scenarios 2) and 3).
Minor comment
Figure 1: Data or Date?
Author Response

(The authors gave the same response as above.)

Round 2
Reviewer 1 Report
Thank you for addressing the comments. The manuscript has been significantly improved.
Author Response
Dear Reviewer,
Thank you very much for reviewing our manuscript entitled "A machine learning framework to predict the occurrence and development of infectious diseases in laying hen farms-taking H9N2 as an example". And thanks for your valuable comments for improving our manuscript.
Sincerely,
Yu Liu.
Reviewer 3 Report
The authors were able to address my comments on the previous version of the manuscript, and the revised manuscript was improved significantly. I would suggest the current form of the manuscript be considered for publication.
However, I disagree with the authors' responses to Comment #7 "...where our five-fold cross-validation performed equally...". In five-fold cross-validation, the training and validation sets were split at random. Even though the five splits covered the entire dataset, there was no guarantee that training-validation splits led to the independence between the two sets. In the split-by-house validation strategy, the training-validation splits could be considered independent in terms of housing units/conditions. So, the two validation strategies cannot be considered equivalent. To conclude, the authors did not mention this argument in the manuscript but should be careful when there's intent.
Author Response
Dear Reviewer,
Thank you very much for reviewing our manuscript entitled "A machine learning framework to predict the occurrence and development of infectious diseases in laying hen farms-taking H9N2 as an example". And thanks for your valuable comments for improving our manuscript.
To our knowledge, the data partition method used in our manuscript is acceptable in machine learning studies. In order to avoid the misunderstanding of its work performance, we have revised the “generalization” and “generalizability” as “good applications ability” and “good applications” in Page 4 line 168 and Page 11 line 362.
Sincerely,
Yu Liu.